# LET'S REWARD STEP BY STEP:
# STEP-LEVEL REWARD MODEL AS THE NAVIGATORS FOR REASONING

## ABSTRACT

Recent years have seen considerable advancements in multi-step reasoning with Large Language Models (LLMs). The previous studies have elucidated the merits of integrating feedback or search mechanisms during model inference to improve the reasoning accuracy. The Process-Supervised Reward Model (PRM), typically furnishes LLMs with step-by-step feedback during the training phase, akin to Proximal Policy Optimization (PPO) or reject sampling. Our objective is to examine the efficacy of PRM in the inference phase to help discern the optimal solution paths for multi-step tasks such as mathematical reasoning and code generation. To this end, we propose a heuristic greedy search algorithm that employs the step-level feedback from PRM to optimize the reasoning pathways explored by LLMs. This tailored PRM demonstrated enhanced results compared to the Chain of Thought (CoT) on mathematical benchmarks like GSM8K and MATH. Additionally, to explore the versatility of our approach, we develop a novel method to automatically generate step-level reward dataset for coding tasks and observed similar improved performance in the code generation tasks. Thus highlighting the robust nature of our reward-model-based approach to inference for reasoning tasks.

## 1 INTRODUCTION

In the exciting evolution of Large Language Models (LLMs) such as GPT (OpenAI, 2023; Brown et al., 2020), LLaMA (Touvron et al., 2023a;b), OPT (Zhang et al., 2022a), Falcon (Penedo et al., 2023), and PaLM (Anil et al., 2023; Chowdhery et al., 2022), a consistent ability to handle tasks from conversation to text generation has been evident. However, when it comes to reasoning, especially multi-step reasoning, current LLMs, even with sophisticated prompting techniques like the Chain of Thought (CoT)(Wei et al., 2023), are still prone to a cascade of errors in their generation processes. As the number of reasoning steps increases, these LLMs face challenges in providing and integrating effective feedback, resulting in one error leading to another.

Achieving a refined multi-step reasoning capability for LLMs can unlock their potential across an even broader array of applications, ranging from complex problem-solving to high-level intellectual tasks. Furthermore, as these models become foundational tools in numerous domains, ensuring their reasoning accuracy is paramount to prevent compounding errors and instill trust in their outputs.

To address the challenges of multi-step reasoning, prior research has focused on feedback mechanisms and supervisory techniques. Works have incorporated self-reflection (Madaan et al., 2023; Xie et al., 2023) or external observation (Yao et al., 2023b; Gou et al., 2023) during reasoning, aiming to correct errors and determine correct reasoning steps. Additionally, novel methods like Tree of Thought (ToT)(Yao et al., 2023a) and RAP(Hao et al., 2023) have integrated traditional search algorithms to convert multi-step reasoning into a path search problem. However, while some of these solutions have led to improvements, they introduce new sets of problems. For instance, the risk of LLMs falling into repetitive thought loops or their self-reflection process exceeding the model's context window constraints. Additionally, direct search processes, such as BFS or DFS, often result in redundancy and substantial reasoning overhead, which, in extreme cases, can degrade to exhaustive search complexity.

Building upon previous studies, it's found that enhancing the reasoning capabilities of large language models involves a few essential aspects. These include step-by-step reasoning, which helps in breaking down complex problems incrementally and guiding the model toward the solution. Another crucial element is the incorporation of feedback or self-reflection. Such feedback can be from the model itself, other models, or even external sources like the results of code execution. Additionally, implementing heuristic search algorithms can be beneficial. Especially when trying to facilitate the model's reasoning path search, it becomes important to compress the search space effectively.

Uesato et al. (2022) and Lightman et al. (2023) introduced the process-supervised reward model (PRM). PRM can provide step-level feedback for the multi-step reasoning result generated by the language model, but it has traditionally been used in RLHF (Ouyang et al., 2022; Luo et al., 2023) via proximal policy optimization (PPO) (Schulman et al., 2017) or reject sampling (Yuan et al., 2023; Dong et al., 2023; Bai et al., 2022). Thus, we hypothesized that the fine-grained feedback from PRM, which enhances RLHF (Wu et al., 2023), could similarly improve reasoning path searching. PRM was used as a reward model for the PPO process in Luo et al. (2023). Ideally, PRM would not introduce additional training overhead and could be further utilized in the decoding phase after the PPO stage to enhance the model's reasoning capabilities. On the other hand, PRM only requires evaluation for each step, making its computational cost much lower than model self-reflection.

Consequently, we propose a heuristic greedy search algorithm for large language model reasoning tasks that harnesses the process-supervised reward model for step-level feedback(HGS-PRM). Similar to the Lightman et al. (2023), we trained a process-supervised reward model based on the open-source model. With our method, each new reasoning step generated by the large language model undergoes evaluation by the reward model. This determines whether to accept the given step or continue generating a new one. If a valid new step is unattainable, a backtrack occurs until a complete reasoning path is identified. Our experimental results on GSM8K (Cobbe et al., 2021) and MATH (Hendrycks et al., 2021) both surpassed those achieved with the CoT method. For instance, with the WizardMath-13B (Luo et al., 2023)model, our method achieved accuracies of 65.4% and 13.7% on the GSM8K and MATH datasets, respectively, surpassing CoT's 63.2% and 10.4%.

Beyond mathematical problems, we aimed to demonstrate the efficacy of HGS-PRM in code generation and even more intricate reasoning tasks. By leveraging technologies like Abstract Syntax Trees (AST) and Mutation Testing, we automatically generated PRM data for code generation challenges on the MBPP (Austin et al., 2021) dataset. Following this, we trained a process-supervised reward model specifically tailored for code. Moreover, our approach also yielded improved results on the HumanEval (Chen et al., 2021) benchmark. For instance, the results for Code-LLaMA-Python-13B increased from 41.5% to 44.5%.

In summary, our contributions are as follows:

1. We explored the feasibility of utilizing PRM during the decoding phase. Training and deploying PRM is a challenging task. By demonstrating the viability of PRM-assisted decoding on open-source models, we've achieved a significant advancement.

2. We evaluated the strengths and weaknesses of prior methods for enhancing large language model reasoning capabilities. Innovatively, we combined PRM with path search, introducing a heuristic greedy search algorithm that leverages PRM for step-level feedback.

3. Ours is the pioneering effort to apply PRM in the coding domain. We have released a PRM dataset specifically for code and validated that PRM is effective across a broader spectrum of reasoning tasks.

## 2 METHOD

In this section, we delve into the technical methodologies we have adopted. First and foremost, we discussed how to use step-level data to train our process-supervised reward model(PRM). Moreover, we provide an in-depth examination of the heuristic greedy search algorithm with PRM (HGS-PRM), discussing its operational mechanics and core concepts. To demonstrate the versatility of this method for other reasoning tasks, We will also introduce how we generated our code-specific PRM data by employing Abstract Syntax Tree (AST) techniques to automate the generation of PRM data tailored for code-related tasks.

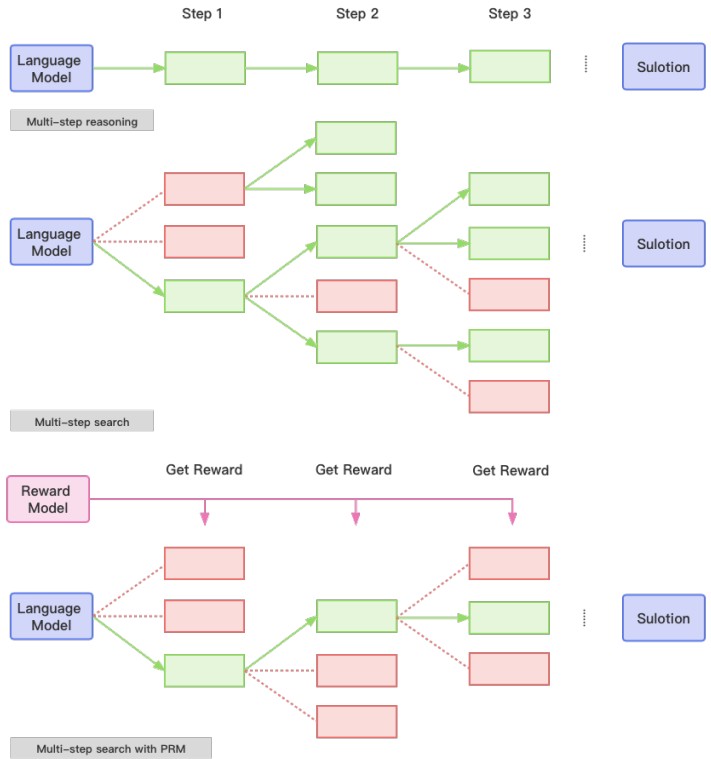

Figure 1: The illustrations of the differences among multi-step reasoning, multi-step search, and multi-step search with PRM

## 2.1 PROCESS-SUPERVISED REWARD MODEL

Following Taori et al. (2023) and Lightman et al. (2023), we trained our reward model, with the entire process divided into two phases:

1. To equip the foundational LLaMA model with mathematical instruction capabilities and strengthen its mathematical proficiency, we initially used the training portion of the MATH dataset. In line with Taori et al. (2023) approach, we mapped 'question' and 'solution' from the MATH(Hendrycks et al., 2021) training dataset to 'instruction' and 'response', respectively. We then performed instruction tuning(Wang et al., 2023c) on the LLaMA-7B(Touvron et al., 2023a) model and got LLaMA-7B-SFT.

2. Building on the instruction-tuning basis, we trained a reward model using PRM800k(Lightman et al., 2023) based on LLaMA-7B-SFT. This model is designed to score each step of the Solution on a step-by-step dimension. The labels fall into three categories: Positive, Negative, and Neutral.

## 2.2 HEURISTIC GREEDY SEARCH WITH PRM

Heuristic search is an algorithmic strategy that utilizes heuristic information in problem-solving to guide the search direction, thereby finding solutions more rapidly. A classic example of heuristic greedy search is the A* algorithm, which combines the benefits of best-first search and Dijkstra's algorithm. It efficiently finds the shortest path by using a heuristic to estimate the cost to the goal from a given node.

During the mathematical problem-solving process with large language models, each chain of reasoning can be broken down step by step. This can be translated into a search problem for the optimal reasoning path within a vast space. Assessing the reward of each step, or the value of a specific

state, is crucial for enhancing the inference capabilities of large language models through search algorithms.

Utilizing the language model's self-evaluation to judge the value of a step can lead to significant computational overhead. For autoregressive models like GPT(Radford et al., 2019) that rely solely on decoding, the computational time complexity is $O(N^2)$. The introduction of self-assessment lengthens the sequence, causing the time complexity to grow exponentially. Additionally, traditional search methods like BFS and DFS have an expansive search space and involve many redundant nodes.

In comparison, using only a process-supervised reward model for assessment significantly reduces overhead compared to self-assessment. Moreover, heuristic greedy search algorithms can effectively minimize the search space, yielding faster results.

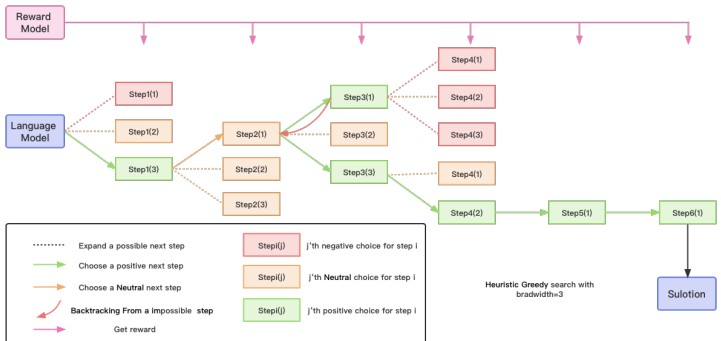

Figure 2: The illustration above depicts the workflow of the heuristic greedy search assisted by the process-supervised reward model, encompassing the expansion of the next step, scoring by the reward model, backtracking, and pinpointing the final result. Red denotes a negative step, Green indicates a positive step, and Orange signifies a neutral step.

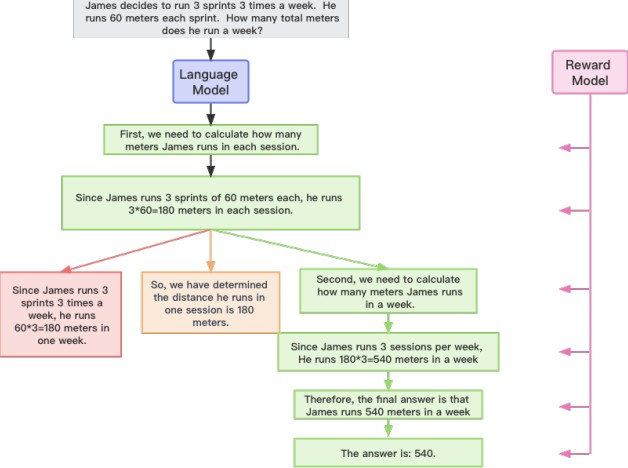

Figure 3: An example of HGS-PRM process. Red denotes a negative step, Green indicates a positive step, and Orange signifies a neutral step.

Consequently, we propose a Heuristic Greedy search algorithm for large language models based on feedback from the process-supervised reward model(Figure 2) as detailed in Appendix A. Figure 3 shows an example of HGS-PRM for a GSM8K problem.

The Algorithm in the AppendixA illustrates the process of our heuristic greedy search algorithm. Given an input question $x$, our algorithm expands to the next potential node and then evaluates it using a reward model. If the reward is positive, the algorithm advances using this new node. If not, it continues to explore the next potential node, assessing it with the reward model until it reaches the maximum bandwidth $B$. If a node expands to the maximum bandwidth without finding a positive step, two scenarios are considered: If neutral sub-nodes exist, one is selected. If all sub-nodes are negative, this indicates an erroneous step. The large language model is unable to determine the correct subsequent step, prompting a backtrack. To constrain our search space efficiently and balance exploration with efficiency, we've set max bandwidth $B$ and max iterations $T$ parameters. Once the algorithm reaches the max iteration count, it will generate an answer from the current step. In our actual implementation, we also tried BFS and MCTS. However, we found that they introduced a more redundant search space and did not effectively improve the results. Our ultimate goal is to enhance the model's mathematical reasoning ability, not to increase algorithmic complexity. Therefore, the heuristic greedy search achieves a better trade-off between exploration and efficiency and is a more optimal choice.

## 2.3 Process-supervised Data for Code

To extend our approach to the field of code, a step-level reward dataset for code is necessary. As opposed to maths problems where it is difficult to assess the correctness of a given reasoning step without human intervention, coding problems with unit-tests are uniquely suited for automated labeling of step-level reward data. In our work, we use MBPP as seed dataset from which our PRM for code dataset is created.

Figure 4 provides an illustration of our method for generating code PRM data. Similar to PRM data where the validity of each reasoning step is evaluated in the context of previous steps, the format of our PRM-Code data code also includes components such as (**PREVIOUS DATA**, **STEP** and **LABEL**). As Python programs are naturally divided into lines separated by the new-line token, we first collect positive results from the ground truth data. For example, if we select the n-th step as the current annotated step, the **PREVIOUS DATA** includes the prompt and steps 0 to n-1, the **STEP** is the n-th step, and the **LABEL** is positive .

The creation of step-level reward data with neutral and negative reward for code can the be accomplished with a combination of code mutation and unit-testing. Automatic code mutation (as in Mutation Testing) is a standard practice in code development where operators (such as addition) in a given code block is automatically mutated to similar operators (such as division). Such mutation requires first transforming a given code block into an abstract syntax tree (AST) representation, and defining rules of how different nodes on the AST are to be mutated. This yields N mutated code variants for each line of code where each mutated code is executable, and each variant undergoes only one mutation. These mutated codes are then executed with unit tests. If a mutated code passes the unit tests, its **LABEL** for the corresponding step is marked as neutral . If it fails to pass the unit tests, the **LABEL** is marked as negative . The **PREVIOUS DATA** includes the prompt and steps 0 to the previous step of the mutation step, the STEP corresponds to the mutation step generated.

We note that mutation testing primarily focus on atomic operators such as arithmetic operations and array slicing, which limits the generality of our PRM-Code dataset. Natural extensions of our current approach involves defining more elaborate mutation rules that involve, for example, mutating for-loops into while-loops that correspond to changes of multiple lines of code. However, despite this limitation, our PRM-Code dataset has proven effectively in improving the code generation performance as we demonstrate in subsequent sections.

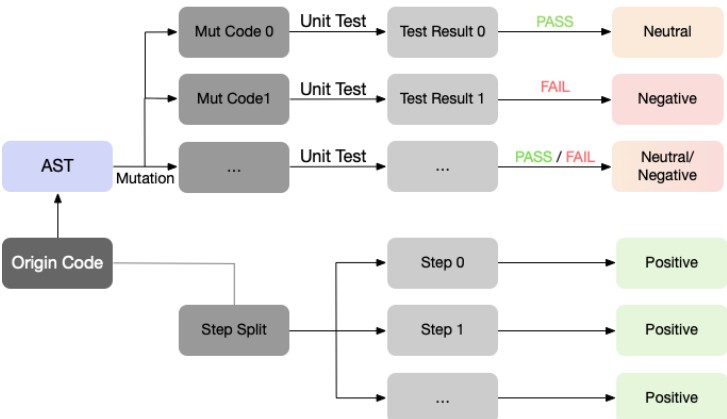

Figure 4: **The process of generating the PRM-Code dataset with ground truth code and unit test.**

## 3 EXPERIMENT

In this section, we will delve into the intricacies of training our processed reward model and present the evaluation results of our heuristic greedy search algorithm, based on this processed reward model, in mathematical and code generation tasks.

### 3.1 PROCESS-SUPERVISED REWARD MODEL TRAINING

For mathematical tasks, we trained our process-supervised reward model(PRM) based on LLaMA-7B(Touvron et al., 2023a). As previously mentioned, our model training method first involved directive fine-tuning using the MATH training set, followed by reward model training. However, it should be noted that we also directly trained our reward model on LLaMA-7B. Our experimental results indicate that models fine-tuned with mathematical directives perform superiorly in all aspects compared to the base model.

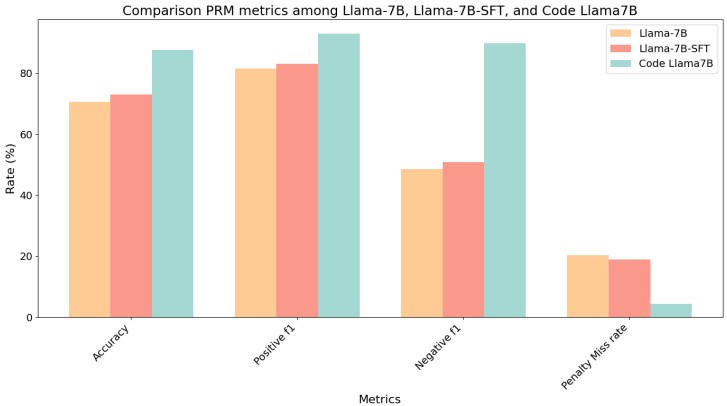

Figure 5: Key metrics of different processed-supervised reward model

From our result shown in Figure 5, the Instruction-tuning model with math data before the training reward model results in improved performance. We also find positive labels exhibit the highest precision and recall. Therefore, we adopted a greedy strategy when designing our search algorithm, directly selecting the step when a positive step is identified. We also find The ability to distinguish

neutral labels is inadequate, so give them less consideration during decoding. They can be treated as either positive or negative.

Penalty miss rate refers to the proportion of classifying positives as negatives and negatives as positives in the mis-classification, which we consider to be the most serious classification error of the reward model. For neutral samples, We can avoid misjudgments by adjusting the action of our search algorithm for neutral samples.

We also trained a process-supervised reward model specifically for code. Drawing from our experiments with math, we directly use Code-LLaMA-7B(Rozière et al., 2023), which had been fine-tuned with code instructions, to train our process-supervised reward model for code. Our code step-level data was automatically generated on MBPP(Austin et al., 2021) with the method in Figure 4.

From our result in Figure 5, the process-supervised reward model(PRM) demonstrated a significantly better ability to discern negative cases in code compared to math. This suggests that there's greater potential in applying the PRM within the coding domain.

## 3.2 MATH RESULT

For mathematical problems, we first conducted evaluation experiments on the general models LLaMA-7B and LLaMA-13B using GSM8K and MATH datasets. We achieved results on both GSM8K and MATH datasets that were superior to those of CoT. For the MATH task, we follow the setting of (Lightman et al., 2023) and evaluate our models only on the 500 MATH test problems because 4.5K MATH test problems are in the PRM800K training set, which we used to train the reward model. Apart from general models, we also conducted experiments on math-specific models WizardMath-7B and WizardMath-13B(Luo et al., 2023). We similarly achieved superior results to CoT on both the GSM8K and MATH datasets. Moreover, we observed that math-specific models benefit more from the reward model. If the language model's intrinsic capability is too weak, even with the aid of a reward model, it remains challenging to sample the correct reasoning path. On the other hand, if the linguistic capacity of the model significantly surpasses that of the reward model, the benefits might not be pronounced. Therefore, aligning the capabilities of the reward model and the language model is of paramount importance.

Table 1: We adopted the sampling setting with temperature=0.1 and top-p=0.95. For LLaMA, we use an 8-shot prompt for GS8K and a 4-shot prompt for the MATH task. For WizardMath, we follow the Luo et al. (2023) prompt

| model | method | GSM8K | MATH |
|---|---|---|---|
| LLaMA2-7B | CoT | 15.7% | 2.4% |
| LLaMA2-7B | HGS-PRM | 16.2%(+0.5%) | 2.6%(+0.2%) |
| LLaMA2-13B | CoT | 31.7% | 2.8% |
| LLaMA2-13B | HGS-PRM | 32.9%(+1.2%) | 4.1%(+1.3%) |
| WizardMath-7B | CoT | 54.3% | 9.7% |
| WizardMath-7B | HGS-PRM | 57.2%(+2.9%) | 11.1%(+1.4%) |
| WizardMath-13B | CoT | 63.2% | 10.4% |
| WizardMath-13B | HGS-PRM | 65.4%(+2.2%) | 13.7%(+3.3%) |

## 3.3 CODE RESULT

To avoid data leakage, our code PRM's step-level training data is created based on MBPP dataset(Austin et al., 2021), and the resulting reward model is evaluated on the HumanEval(Chen et al., 2021) benchmark. HumanEval is a widely recognized benchmark for testing large language models on coding tasks. It consists of 164 original programming questions, with an average of 9.6 test cases assigned to each question.

Table 2: We follow the previous works Chen et al. (2021) to estimate the pass@1 score with the same set of hyper-parameters: temperate=0.2, and top-p=0.95

| model | method | HumanEval pass@1 |
|-------|--------|------------------|
| Code-LLaMA-Python-7B | CoT | 36.6% |
| Code-LLaMA-Python-7B | HGS-PRM | 41.5%(+4.9%) |
| Code-LLaMA-Python-13B | CoT | 41.5% |
| Code-LLaMA-Python-13B | HGS-PRM | 44.5%(+3.0%) |

For our evaluation, we selected models Code-LLaMA-Python-7B and Code-LLaMA-Python-13B. On the HumanEval benchmark, our approach improved pass@1 result by 4.9%(36.6% VS 41.5%) for Code-LLaMA-Python-7B model and 3.0% (41.5% VS 44.5%) for Code-LLaMA-Python-13B model over CoT, respectively. This performance also surpassed the greedy decoding results presented in the Code-LLaMA paper(Rozière et al., 2023), which is 3.1%(38.4% VS 41.5%) and 1.2%(43.3% VS 44.5%). The accuracy improvement we observed in code is generally higher than in mathematics. This aligns with our prior evaluation results during the PRM training reward phase where the precision and recall for the Code PRM negative label were notably higher than for math. We hypothesize that this might be because both HumanEval and MBPP involve relatively simple programming challenges, whereas MATH presents more complex mathematical problems which are intrinsically more challenging for both PRM and the language models themselves to learn.

## 3.4 FURTHER ANALYSIS

Is the result obtained through HGS-PRM reliable? We further analyzed the relationship between PRM feedback and the correctness of the results. The actual process is often not so ideal. As indicated in 5, the actual precision of PRM is lower than 90%. If the accuracy of PRM is not up to par, HGS-PRM might reduce the accuracy of the results. Therefore, besides the greedy search method, we can also use the approach of scoring after sampling to verify whether PRM is effective.

For the field of mathematics, we sampled 10 times from the MATH dataset with LLaMA2-13B and then scored the sampled reasoning paths at the step-level using PRM. According to our HGS-PRM algorithm, the correct reasoning path should not have any negative steps, so we filtering all samples with negative steps. For code, we sampled 100 times from the HumanEval dataset using Star-Coder(Li et al., 2023) and scored in the same manner.

From our results in Table 3, the "Accuracy after filtering"(14.4%, 36.66%) is much higher than the original accuracy(4.25%, 30.13%), demonstrating the correlation between the step-level feedback provided by PRM and the correctness of the result. The computational cost of filtering after repeated sampling is higher than that of HGS-PRM, because, for complex reasoning problems, it's challenging for large language models to sample reasoning paths that don't contain negative steps. This is also why we use search instead of sampling.

Table 3: We adopted the sampling setting with temperature=0.8 and top-p=0.95 for reasoning path sampling using LLaMA2-13B model. The **Accuracy after filtering** refers to the accuracy of sampled responses filtered by PRM model that contain no negative reasoning steps.

| | MATH | HumanEval |
|---|------|-----------|
| **Accuracy** | 4.25% | 30.13% |
| **Accuracy after filtering** | 14.4% | 36.66% |

## 3.5 CASE STUDY

Appendix B displays some correct reasoning paths and erroneous paths obtained by HGS-PRM, further detailing what mistakes PRM specifically identified such as parsing errors, computational errors, and semantic errors.

## 4 RELATED WORKS

**Large Language Models For reasoning** Large language models (LLMs) have faced challenges in complex reasoning tasks, covering mathematical reasoning (Lu et al., 2023; Frieder et al., 2023) and code generation (Chen et al., 2021; Li et al., 2023; Rozière et al., 2023). Key datasets for mathematical reasoning are GSM8K (Cobbe et al., 2021), MATH (Hendrycks et al., 2021), AddSub (Hosseini et al., 2014), MultiArith (Roy & Roth, 2015), and SingleEQ (Koncel-Kedziorski et al., 2015) while code generation evaluations employ HumanEval (Chen et al., 2021), MBPP (Austin et al., 2021), and DS1000 (Lai et al., 2022).. Wei et al. (2023) proposed "Chain-of-Thought Prompting" (CoT) to improve LLMs' performance by prompting step-by-step decompositions. The Zero-shot-CoT by Kojima et al. (2023) enhances this by adding 'Let's think step by step' to responses, detailing the model's reasoning. Gao et al. (2023) introduced the Program-aided Language Model (PAL) to assist LLMs in reasoning, while Wang et al. (2023b) presented a self-consistent method building upon CoT for consistent responses. Zhao et al. (2023) combined both CoT and PAL for improved reasoning, allowing concurrent use with the self-consistent approach. Lastly, Zhang et al. (2022b)'s Auto-CoT leverages LLMs to produce diverse reasoning chains autonomously.

**Reasoning with feedback** Beyond prompt-based enhancements, Pan et al. (2023) reviews methods combining self-feedback, search, and tools (Gou et al., 2023) to amplify large language model reasoning. Research by Madaan et al. (2023), Gero et al. (2023), Chen et al. (2023), Weng et al. (2023) and Shinn et al. (2023) demonstrate that these models can serve as feedback sources to bolster reasoning. Xie et al. (2023) offers a prompting method integrating self-evaluation via stochastic beam search to refine multi-step inference. Yao et al. (2023a)'s "ToT" (Tree of Thoughts) advances CoT, enabling models to explore varied reasoning paths, evaluate decisions, and adjust strategies. Hao et al. (2023) transforms the LLM to act as both a world model and logical agent, using a planning method inspired by Monto Carlo Tree Search. Wang et al. (2023a)'s LeTI uses textual feedback from code errors for better code generation. Yang et al. (2023) takes a retrieval approach, allowing LLMs to extract data from Lean and engage with proof environments. Finally, Yao et al. (2023b) presents a framework blending advances in reasoning and action for a broader spectrum of linguistic challenges.

### PRM & ORM

While much attention has been given to other areas, the process-supervised reward model (PRM) and outcome-supervised reward model (ORM) have seen less exploration. Uesato et al. (2022) first introduced PRM, highlighting its advantages over ORM in several applications, from few-shot prompting to reward modeling. Expanding on this, Lightman et al. (2023) released PRM800K, a dataset based on MATH annotations, showcasing the reliability of process supervision over outcome supervision. This high-quality dataset has been invaluable to our research. Luo et al. (2023) introduced "Reinforcement Learning from Evol-Instruct Feedback (RLEIF)", using PRM as a reward model within the PPO framework (Schulman et al., 2017). While these studies have focused on PRM for math, there's a noticeable gap in PRM research for coding, pointing to a ripe area for further investigation.

## 5 CONCLUSIONS AND FUTURE WORK

In this study, we introduce an approach that integrates the process-supervised reward Model (PRM) into heuristic greedy search, demonstrating the value of incorporating PRM feedback in multi-step inference tasks within large language models. This offers fresh insights into utilizing the PRM framework rather than RLHF or reject sampling. Our experimental results on the MATH and GSM8K datasets validate the effectiveness of our proposed HGS-PRM method and explore the potential of PRM in the decoding phase experiment.

Moreover, we take advantage of the Mutating Testing technique for code PRM data generation, enabling rapid scaling of step-level reward data for code applications. We release a PRM dataset tailored for coding, which holds pivotal significance for further research on PRM's application in code-related tasks. We believe that the potential of the process-supervised reward model in multi-step reasoning within expansive models remains under-explored, especially in coding tasks. Enhancing code-specific PRM data and reward models could significantly bolster the capabilities of large language models in code generation.

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

## A    HEURISTIC GREEDY SEARCH WITH PRM(HGS-PRM)

1. **Node definition** Each node of the search tree possesses the following elements.

   (a) **State** $S = [x, s_1...s_i]$, where the input prompt and question $Q$ represented as $x$, each step $i$ represented as $s_i$

   (b) **Step** $s_i$, the reasoning step for this node

   (c) **Reward** $r$, the reward for the step $i(s_i)$ evaluated from the PRM

   (d) **Value** $V$, the cumulative reward for each step leading to that node.

   (e) **Parent** $parent$, the parent of the node

   (f) **Children** $children$, the children node of the node

2. **Expand** Expand the possible next steps from the current node, and create a new child node:$P(s_{i+1}|S) = P(s_{i+1}|x, s_1 s_i)$. the sampling diversity is influenced by the sample hyper-parameter like temperature, top-$p$, top-$k$.

3. **Get Reward**: after expanding a new node, the process reward model will evaluate the new step and feedback reward range from $[-1, 0, 1]$, which means `negative`, `neutral`, `positive` step.

4. **Go ahead**: when the reward of the step is `positive` (r = 1), we will choose this step and go ahead to this new child node.

5. **Backup**: When this node cannot expand into different steps or the number of child nodes reaches the maximum bandwidth, it will backtrack to its parent node.

---

**Algorithm 1** Heuristic Greedy Search with PRM(HGS-PRM)

---

**Require:** Input Question $x$, LLM generate function$P()$, next step Expand function E(), get reward function $R()$, max bandwidth $B$, max iterations $T$, end token or end sequence of LLM $eos\_token$

1: $cur\_node \leftarrow \{x\}$
2: **for** $i = 1$ to $T$ **do**
3:     **if** number of $cur\_node.child < B$ **then**
4:         $next\_node \leftarrow E(P, cur\_node)$
5:         **if** $eos\_token$ in $next\_node.step$ **then**
6:             **return** $next\_node.state$
7:         **else if** $next\_node.reward == 1$ **then**
8:             $cur\_node \leftarrow next\_node$
9:         **end if**
10:     **else**
11:         **if** $child.reward = 0$ for any child in $cur\_node.children$ **then**
12:             $cur\_node \leftarrow$ one of the child
13:         **else**
14:             $cur\_node \leftarrow cur\_node.parent$
15:         **end if**
16:     **end if**
17: **end for**
18: $result \leftarrow P(cur\_node.state)$
19: **return** $result$

---

# B CASE STUDY

## B.1 MATH CASE STUDY

| | |
|---|---|
| **Question** | Riproarin' Ringo was roping a recalcitrant dogie. Ringo decided to give the dogie a reprieve by calculating $$\|(1-i)^8\|$$ before riding after the dogie. What answer should Ringo have found? |
| **Ground Truth** | **540** |
| **Correct path** | We are given a complex number in the form of $(1-i)$, and we need to find the magnitude of its eightfold, which is $(1-i)^8$. |
| | The magnitude of a product is the product of the magnitudes, i.e., $\|ab\| = \|a\|\|b\|$. In our case, we have $(1-i)^8$, and we want to find $\|(1-i)^8\|$. |
| | Using the property from Step 2, we can rewrite $\|(1-i)^8\|$ as $\|1\|\|(1-i)\|^8\|1\|$. Since the magnitude of a number is always non-zero, we can safely cancel out the $\|1\|$ terms, leaving us with $\|(1-i)\|^8$. |
| | Now, we need to find the magnitude of $(1-i)$. The magnitude of a complex number $a+bi$ is given by $\sqrt{a^2+b^2}$. In our case, $a=1$ and $b=-1$, so the magnitude of $(1-i)$ is $\sqrt{1^2+(-1)^2}=\sqrt{2}$. |
| | Finally, we need to find $\|(1-i)^8\|$. Since we found that $\|(1-i)\|=\sqrt{2}$, we can raise it to the power of 8: $(\sqrt{2})^8$. |
| | We can simplify $(\sqrt{2})^8$ as $2^{\frac{8}{2}}$. Since $\frac{8}{2}$ is equal to 4, we have $2^4=16$. |
| | So, the final answer is $\boxed{16}$. |
| | The answer is: 16. |
| **Wrong path (computational errors)** | We are given a complex number in the form of $(1-i)$, and we need to find the magnitude of its eightfold, which is $(1-i)^8$. |
| | Before finding the magnitude of $(1-i)^8$, we need to find the magnitude of $(1-i)^2$. To do this, we can use the formula for the square of a complex number: $(a+bi)^2=a^2+2abi+b^2$. In our case, $a=1$ and $b=-1$, so $(1-i)^2=(1)^2+2(1)(-1)+(-1)^2=1-2i+1=2-2i$. |
| | Now, we need to find the magnitude of $(2-2i)^4$. Using the same formula for the square of a complex number, we get $(2-2i)^4=(2)^4+2(2)(-2)i+(-2)^4=16-32i+16=32-32i$. |
| | Next, we need to find the magnitude of $(32-32i)^2$. Using the formula for the square of a complex number again, we get $(32-32i)^2=(32)^2+2(32)(-32)i+(-32)^2=1024-2048i+1024=2048-2048i$. |
| | Finally, we need to find the magnitude of $(2048-2048i)^4$. Using the formula for the square of a complex number one more time, we get $(2048-2048i)^4=(2048)^4+2(2048)(-2048)i+(-2048)^4=16,777,216-4,195,392i+16,777,216=33,554,508$. Now, we take the absolute value of the result: $\|33,554,508\|=33,554,508$. So, the final answer is $\boxed{33,554,508}$. The answer is: 33,554,508. |

Table 4: MATH case with WizardMath-13B

## B.2   GSM8K CASE STUDY

| Question | James decides to run 3 sprints 3 times a week. He runs 60 meters each sprint. How many total meters does he run a week? |
|---|---|
| **Ground Truth** | **540** |
| **Correct path** | First, we need to calculate how many meters James runs in each session. |
| | Since he runs 3 sprints of 60 meters each, we can calculate the total distance he runs in each session by multiplying the number of sprints by the distance of each sprint. |
| | Therefore, James runs 3*60=180 meters in each session. |
| | Second, we need to calculate how many meters James runs in a week. |
| | Since he runs 3 sessions per week, we can calculate the total distance he runs in a week by multiplying the distance he runs in each session by the number of sessions he runs per week. |
| | Therefore, James runs 180*3=540 meters in a week. |
| | Therefore, the final answer is that James runs 540 meters in a week. |
| | The answer is: 540. |
| **Wrong path** (parsing errors) | First, we need to calculate how many meters James runs in one sprint. |
| | According to the question, James runs 60 meters each sprint. |
| | Therefore, we can calculate the total distance he runs in one week by multiplying the distance of one sprint by the number of sprints he runs in one week. |
| | Since James runs 3 sprints 3 times a week, he runs 60*3=180 meters in one week. |
| | Therefore, the total distance James runs in one week is 180 meters. Thus, the final answer is 180. The answer is: 180. |

Table 5: GSM8K case with WizardMath-13B

### B.3 HUMANEVAL CASE STUDY

| Prompt | |
|---|---|
| | ```
def how_many_times(string: str, substring: str) -> int:
    """ Find how many times a given substring
    can be found in the original string.
    Count overlaping cases.
    >>> how_many_times('', 'a')
    0
    >>> how_many_times('aaa', 'a')
    3
    >>> how_many_times('aaaa', 'aa')
    3
    """
``` |
| **Correct path** | |
| | ```
if len(substring) == 0:
    return 0
if len(string) == 0:
    return 0
if len(string) < len(substring):
    return 0
count = 0
for i in range(len(string) - len(substring) + 1):
    if string[i:i + len(substring)] == substring:
        count += 1
return count
``` |
| **Wrong path** (semantic errors) | |
| | ```
if len(substring) == 0:
    return 0
return string.replace(substring, '', 1).count(substring)
``` |

Table 6: HumanEval case with Code-LLaMA-Python-13B

