# OpenReview forum: "Let's reward step by step: Step-Level reward model as the Navigators for  Reasoning"
_ICLR.cc/2024/Conference — ICLR 2024 Conference Withdrawn Submission_

### Official Review · Reviewer_A78C · 2023-10-29

**Soundness:** 2 fair
**Presentation:** 2 fair
**Contribution:** 1 poor
**Rating:** 3
**Confidence:** 4

**Summary:**

This paper proposes the math- / code-specialized process supervised reward model (PRM) for large language models' reasoning. By finetuning LLaMA-7B (SFT/Code variants) on PRM800K dataset for math and the generated code dataset based on MBPP, PRM are trained for specific reasoning problems. The dataset for code is generated via the Mutation Testing process. The method mainly choose the positive-label reasoning node, and if the reward labels of the child nodes predicted by PRM are all the negative ones, the process is backtracked. Such PRMs improve the accuracy of mathematical reasoning of LLaMA2-7/13B and WizardMath-7/13B and HumanEval pass@1 of Code-LLaMA-Python-7/13B compared to Chain-of-Thought prompting.

**Strengths:**

### significance
- PRM can be trained with tractable-size LLMs (7 billion parameters).

### clarity
- The data generation process for coding experiments is clearly described.

**Weaknesses:**

- Step-wise verification is well-studied in the previous literature (for instance, [1, 2, 3]). The experimental comparison has not been conducted, and I'm not sure what the novel contribution of this paper is.
- The improvement of the performance seems marginal in all the settings (math/code/models). Considering the inference latency, the proposed HGS-PRM might not be a competitive choice.
- The difference between Figure 2 and Figure 3 is unclear. They seem to describe the same procedure.

[1] https://arxiv.org/abs/2305.10601

[2] https://arxiv.org/abs/2305.14992

[3] https://arxiv.org/abs/2305.20050


(Style Issues)
- Only the caption of Figure 4 is bold. I'm not sure it's the intention.
- (In the caption of Table 1) GS8K --> GSM8K, missing colon at the end of sentence.
- It would be good to be consistent in spacing around parentheses (citation, numbers, etc).
- (In Section 4.3) "indicasted in 5" -->  "indicasted in Table/Figure 5"?

**Questions:**

- Does this PRM work with more capable models such as LLaMA2-70B, GPT-3.5-turbo, GPT-4, etc?
- Is there any reason why you use different temperatures (0.1 for math, 0.2 for coding)?
- Is there any reason why you use LLaMA variants for the base LLM of PRM, rather than LLaMA2-7B/WizardMath-7B for math problems?
- In Section 3.4, you seemed to employ Star-Corder, rather than Code-LLaMA-Python. Is there any reason?

---

> ### Author Response · Authors · 2023-11-22
> **Reply**
>
> We thank the reviewer for the comment. We plan on extensively revising our manuscript with more extensive benchmarking and will be withdrawing our submission as a result.
> However, we'd like to address a few points raised by the reviewer for clarification.
> Questions:
> - Does this PRM work with more capable models such as LLaMA2-70B, GPT-3.5-turbo, GPT-4, etc?
>
> Yes, we believe that this approach is viable. However, as mentioned in the results section of our paper, our PRM requires stronger capabilities to effectively match the policy model. This necessitates a balance between the abilities of the PRM and the base model. Additionally, it's not yet clear whether there's a similar scaling law between PRM and the base model. We consider this an intriguing point worth exploring further. Such an investigation could provide valuable insights into optimizing the effectiveness of PRM in various applications.
>
> - Is there any reason why you use different temperatures (0.1 for math, 0.2 for coding)?
>
> For the configuration of our mathematical sampling, we followed the guidelines outlined in https://arxiv.org/abs/2308.09583. For the code aspect, we based our approach on the methods described in https://arxiv.org/abs/2107.03374. The results from this paper indicated that the best outcomes for pass@1 were achieved with a temperature setting of 0.2, so we adopted this setting in our experiments. This decision was made to align with the proven effective practices in these prior studies, providing a solid foundation for our own research.
>
> - Is there any reason why you use LLaMA variants for the base LLM of PRM, rather than LLaMA2-7B/WizardMath-7B for math problems?
>
> Since WizardMath-7B was released on August 18th, we were unable to utilize it for training our PRM before our submission deadline. Based on our experience, PRMs trained on stronger mathematical models tend to perform better. This implies that if we had access to WizardMath-7B earlier, it could have potentially enhanced the effectiveness of our PRM. Future experiments incorporating WizardMath-7B for PRM training are expected to yield more promising results.
>
> - In Section 3.4, you seemed to employ Star-Corder, rather than Code-LLaMA-Python. Is there any reason?
>
> Similarly to the previous question, our initial experiments were conducted on StarCoder until the release of code llama, at which point we found that code llama performed better. We apologize for not having had the time to validate this earlier. This highlights the dynamic nature of research where advancements in models can significantly impact experimental outcomes, and we aim to further investigate and validate these improvements in future work.

---

> ### Comment · Reviewer_A78C · 2023-11-22
>
> Thank you for the response. I surely checked the response. For your interest, I think using the same model for reward models would be better for consistency, otherwise, the reviewer faces confusion.

---

### Official Review · Reviewer_abh9 · 2023-10-31

**Soundness:** 3 good
**Presentation:** 3 good
**Contribution:** 3 good
**Rating:** 5
**Confidence:** 5

**Summary:**

The paper treated the reasoning tasks as a step by step generation task. They interpreted each step as a node of a tree and translated the problem as a tree search problem. They then proposed a greedy search algorithm w/ the similar philosophy as A*; using a trained process reward model (PRM) to provide the signal of values. They also proposed a novel method to generate synthetic training dataset for PRM for coding tasks. The proposed PRM-augmented searching method outperform chain-of-thought baselines on math and coding tasks using some LLaMA-based small models.

**Strengths:**

1. PRM is a good method and I am very happy to see more exploration of its usage. This paper provide more evidence of the effectiveness of PRM.
2. The way of creating synthetic PRM training dataset for coding tasks is very cleaver! It's quite simple very looks effective. The method is very inspiring.
3. Some detailed discussion in the paper is also helpful, e.g., when policy model is way stronger than reward model or vice versa would lead to suboptimal results.

**Weaknesses:**

1. The novelty of the idea can be a possible weakness. The process supervision is not a new thing as both DeepMind and OpenAI have solid studies on math tasks --- the authors also mentioned this. The search algorithm is very similar to the philosophy of A*; searching reasoning paths as tree is also not a new thing (Tree of Thought; or even AlphaGo). So IMO, the novelty of the paper is kinda near the threshold, and I personally tend to below the line. While I do accept different opinions on this as most of the LLM papers nowadays looks quite incremental; and this one is better than those --- the question is what the bar is for ICLR. I'd like to refer to opinions from other reviewers as well.
2. There is no space between parentheses and the proceeding letter in many places in the paper.
3. A lot of details are missing or unclear. Please refer to the questions below.

**Questions:**

1. In Section 2.1 you mentioned that you trained the base model like Alpaca. If so, when generating each node (step), you still need to generate the whole path to the end of the solution, is that correct? If so, that will introduce many extra cost if some early steps are "negative" as the model will continue generation till the end anyway. Please correct me if I understand incorrectly. I didn't see any discussions about this in the paper and this is my largest concern about the efficiency of the search algorithm.
2. How did you determine what a step is for math tasks? By "\n"? IIUC, you didn't conduct the style alignment as Lightman in the PRM paper. Without this step, it is not guaranteed that each step would be separated by "\n".
3. Did you compare your method w/ the sampling and ranking method in the PRM paper? Section 3.4 seems to mention something related but described very unclear. Beating CoT baseline is as expected since you introduced the extra reward model; whether your method can beat other PRM-augmented ranking/search is more important.
4. Did you train your PRM as a classification model? If so, why not train it to produce a continuous value like the PRM paper?
5. In Appendix B.3, the first line of correct and incorrect solution are the same. Why one is positive and the other is neutral?
6. As your method doesn't require to tune the policy model, it is possible to use OpenAI's models as policy models. Did you try it and find it not working since the policy model is much stronger than the reward model?

---

> ### Author Response · Authors · 2023-11-22
> **Reply**
>
> We thank the reviewer for the comment. We plan on extensively revising our manuscript with more extensive benchmarking and will be withdrawing our submission as a result.
> However, we'd like to address a few points raised by the reviewer for clarification.
>
> 1. In Section 2.1 you mentioned that you trained the base model like Alpaca. If so, when generating each node (step), you still need to generate the whole path to the end of the solution, is that correct? If so, that will introduce many extra cost if some early steps are "negative" as the model will continue generation till the end anyway. Please correct me if I understand incorrectly. I didn't see any discussions about this in the paper and this is my largest concern about the efficiency of the search algorithm.
>
> Indeed, the sampling efficiency is a potential downside of our decoding technique, and we also took this issue into account when designing our algorithm, which is why we set a maximum number of search iterations, denoted as \( T \). This approach helps in balancing the computational cost and the effectiveness of the algorithm.
>
> 2. How did you determine what a step is for math tasks? By "\n"? IIUC, you didn't conduct the style alignment as Lightman in the PRM paper. Without this step, it is not guaranteed that each step would be separated by "\n".
>
> Yes, currently our method divides steps based on the '\n' character, which is a relatively universal approach. However, for models like Wizard Math that are fine-tuned with specific instructions, it's also feasible to divide according to the format of Supervised Fine-Tuning (SFT) instructions. This allows for more tailored processing in line with the specific characteristics of the fine-tuned model.
>
> 3. Did you compare your method w/ the sampling and ranking method in the PRM paper? Section 3.4 seems to mention something related but described very unclear. Beating CoT baseline is as expected since you introduced the extra reward model; whether your method can beat other PRM-augmented ranking/search is more important.
>
> Yes, we agree with this statement. Due to the limitations of the base model's parameter size, our current PRM results are not as robust as we would like. We plan to conduct additional experiments in the future to further strengthen our findings and demonstrate the effectiveness of PRM under different conditions and with potentially larger model parameters.
>
> 4. Did you train your PRM as a classification model? If so, why not train it to produce a continuous value like the PRM paper?
>
> Yes, it's a classification model. This is because the dataset from https://github.com/openai/prm800k is annotated with three categories: -1, 0, and 1. This categorization naturally leads to the development of a classification model to handle this type of structured data. However, a continuously valued PRM model is certainly an addition we'll explore in the revised manuscript.
>
> 5. In Appendix B.3, the first line of correct and incorrect solution are the same. Why one is positive and the other is neutral?
>
> Sorry, this was a mistake in our writing. Thank you for pointing it out.
>
> 6. As your method doesn't require to tune the policy model, it is possible to use OpenAI's models as policy models. Did you try it and find it not working since the policy model is much stronger than the reward model?
>
> Indeed, GPT-4's capabilities far exceed those of our PRM model. Our 7B Llama PRM does not have the capacity to comprehend GPT-4, as we mentioned in the results section of our paper, where we discussed the need for the policy model and PRM model capabilities to be aligned. However, from our perspective, PRM is meaningful. It could enhance the inference abilities of some models that are not as powerful as GPT-4, making them more effective in certain scenarios.

---

> > ### Comment · Reviewer_abh9 · 2023-11-22
> >
> > Thanks for the reply and hope you get better results for next submission.

---

### Official Review · Reviewer_X471 · 2023-11-01

**Soundness:** 2 fair
**Presentation:** 2 fair
**Contribution:** 2 fair
**Rating:** 5
**Confidence:** 3

**Summary:**

The submission presents a technique of using PRM to guide decoding for math and coding tasks. The idea is very interesting, and the writing is relatively easy to follow. The experiment results, however, are not super convincing and there are many open questions left. I encourage the authors to conduct more experiments and continue this line of very interesting work.

**Strengths:**

- The idea of using the PRM to guide the reasoning path generation makes a lot of sense. The greedy algorithm also is suitable here for simplicity and for potential efficiency over some current complicated prompting frameworks.
- Generating the code dataset with ground truth code and unit test is also a clever way of synthesizing PRM data, which is very costly to collect

**Weaknesses:**

- I find some of the claims over-generalized and unjustified. For example: “If the language model’s intrinsic capability is too weak, even with the aid of a reward model, it remains challenging to sample the correct reasoning path. On the other hand, if the linguistic capacity of the model significantly surpasses that of the reward model, the benefits might not be pronounced. Therefore, aligning the capabilities of the reward model and the language model is of paramount importance.”
What does the intrinsic capability refers to here? If it’s parameter size, then WizardMath-7B seems to have more improvement on GSM8K tasks than WizardMath-13B. If it’s math specific abilities, then it is not consistent with the claims above.
- “We hypothesize that this might be because both HumanEval and MBPP involve relatively simple programming challenges, whereas MATH presents more complex mathematical problems which are intrinsically more challenging for both PRM and the language models themselves to learn.”: Are there any justifications for such a hypothesis?
- Results: +0.2% of 500 examples is 1 example. In the MATH results. And 0.5% of 1K test examples is 5 examples. Are these within the noise range of the metric?
- RLHF results missing. If we are using Reward Models, another important baseline is the model after RLHF.

*Writing Feedback*

- I find figure 5 a bit confusing because it seems to have two models for MATH, and one for the code task. “ As previously mentioned, our model training method first involved directive fine-tuning using the MATH training set, followed by reward model training. However, it should be noted that we also directly trained our reward model on LLaMA-7B. Our experimental results indicate that models fine-tuned with mathematical directives perform superiorly in all aspects compared to the base model.” – I am then confused as to which model is used for the reward model in the end. If the SFT model performs better, why is the reward model directly trained on LLaMA-7B?
- For rigor, should also report the base mode’s performance on code task.

**Questions:**

N/A

---

> ### Author Response · Authors · 2023-11-22
> **Reply**
>
> We thank the reviewer for the comment. We plan on extensively revising our manuscript with more extensive benchmarking and will be withdrawing our submission as a result.
> However, we'd like to address a few points raised by the reviewer for clarification.
>
> - I find figure 5 a bit confusing because it seems to have two models for MATH, and one for the code task. “ As previously mentioned, our model training method first involved directive fine-tuning using the MATH training set, followed by reward model training. However, it should be noted that we also directly trained our reward model on LLaMA-7B. Our experimental results indicate that models fine-tuned with mathematical directives perform superiorly in all aspects compared to the base model.” – I am then confused as to which model is used for the reward model in the end. If the SFT model performs better, why is the reward model directly trained on LLaMA-7B?
>
> Thanks for your feedback. We have experimented with training PRM using both a Llama model that has undergone Supervised Fine-Tuning (SFT) and one that has not. The results from the latter were not as promising, so we opted for the former approach. During the training of PRM, we encountered numerous challenges; PRM is quite difficult to train, which may be reflected in the experimental results not being as robust as desired. However, we believe in the immense potential of PRM. A well-trained PRM can significantly enhance the inference capabilities of Large Language Models (LLMs). Moreover, due to limitations in GPU scale, we were unable to train PRM on larger-parameter base models.
>
> - For rigor, should also report the base mode’s performance on code task.
>
> For code-related tasks, our own dataset is not as large as prm800k, and the quality of our data does not match that of prm800k. This may result in our experimental outcomes not being as strong. However, we believe that this technological approach is workable. We plan to supplement our work with more robust experimental results in the future.

---

### Official Review · Reviewer_M8mk · 2023-11-02

**Soundness:** 3 good
**Presentation:** 1 poor
**Contribution:** 2 fair
**Rating:** 5
**Confidence:** 3

**Summary:**

Process-supervised reward models (PRMs) provide supervision of whether each step of reasoning is valid. Existing work uses such reward models for fine-tuning a LLM, e.g., via RLHF. Instead, this work proposes to directly leverage PRMs during decoding via a heuristic backtracking algorithm. At decoding time, output is sampled from the language model and evaluated under the PRM. If the PRM feedback is negative, the output is re-sampled (i.e., backtracking), whereas if the feedback is positive, the language model continues to output from there. The results indicate that that this yields improvements over Chain-of-Thought prompting on GSM8K

**Strengths:**

This work proposes a simple and reasonable approach for incorporating PRMs directly into decoding without the need for fine-tuning on them. The reported results are encouraging, and it seems like this work would be interesting to the community and warrant further investigation.

**Weaknesses:**

The main weakness of this work is its presentation, which I do not think is ready for publication. The writing is vague almost everywhere (e.g., lacking a formal description of the proposed approach), which makes it difficult to understand and reproduce the proposed approach.  I think the general ideas behind the paper seem solid and interesting enough, but the presentation needs to be significantly improved for this to be fully appreciated by the community.

**Questions:**

Can the authors provide a precise formal overview of the proposed decoding algorithm and the training procedure for the PRM?

---

> ### Author Response · Authors · 2023-11-22
> **Reply**
>
> We thank the reviewer for the comment. We plan on extensively revising our manuscript with more extensive benchmarking and will be withdrawing our submission as a result.
> However, we'd like to address a few points raised by the reviewer for clarification.
>
> For Questions:
> Can the authors provide a precise formal overview of the proposed decoding algorithm and the training procedure for the PRM?
>
> Thank you for your suggestion. We plan to open our work for future development, ensuring that all training and inference processes are reproducible. We believe PRM has tremendous potential in the domain of code. The training process of PRM is clearly detailed in our section 2.1 PROCESS-SUPERVISED REWARD MODEL, where we describe the dataset used (https://github.com/openai/prm800k) and the Llama2 model. Detailed explanations are also provided in Appendix A, HEURISTIC GREEDY SEARCH WITH PRM (HGS-PRM).

---

### Official Review · Reviewer_Q37j · 2023-11-06

**Soundness:** 2 fair
**Presentation:** 2 fair
**Contribution:** 2 fair
**Rating:** 5
**Confidence:** 2

**Summary:**

As the performance of LLM continues to improve, their ability to do multi-step reasoning is become more important. Currently, most LLM ability to do multi-step reasoning suffers from cascading errors. To address these issues, the authors propose a greedy heuristic search algorithm that performs step-level feedback using PRM to improve LLM multi-step reasoning.

**Strengths:**

Improving multi-step reasoning in LLM is a very important topic. The strengths of this paper are

1. Solution Simplicity: The authors proposed a very simple method with empirical performance superior to the paper's baseline methods.
2. Combination of PRM, Code, and Mutation testing: To perform experiments with PRM, it usually requires a lot of human annotation. However, the observation that PRM can be trained with mutation testing, which provides automatic code atomic code changes and the fail and pass, was creative.

**Weaknesses:**

Though this paper addresses an import problem and has strengths, but there are also some weaknesses outlined below:

1. Lack of baseline: The authors do not compare to common decoding strategies used: majority voting (self-consistency) [1] and RM-weighted decoding (verifier voting) [2].
2. Writing Quality: There are several typos throughout the paper and the paper lacks clarity. Some typos are "We also find The ability to distinguish ...", "directive fine-tuning ...", and "mathematical directives perform...".
3. The idea to sample greedy from the model and score it with the reward function makes strong assumptions on the reward model and starting model abilities.

[1] Self-consistency improves chain of thought reasoning in language models by Wand et al. 2022
[2] Solving math word problems with process- and outcome-based feedback by Uesato et al. 2022

**Questions:**

1. How does the proposed approach compare to majority voting and RM-weighted decoding? Given that PRM has not been used in the code domain - showing the performance of these baselines is important.
2. How does the proposed approach compare to outcome-supervised reward models (ORMs)?
3. Why is self-assessment more expensive than PRM, given that both the PRM and generator use the same LLM?

---

> ### Author Response · Authors · 2023-11-22
> **Reply**
>
> We thank the reviewer for the comment. We plan on extensively revising our manuscript with more extensive benchmarking and will be withdrawing our submission as a result.
> However, we'd like to address a few points raised by the reviewer for clarification.
>
> 1. How does the proposed approach compare to majority voting and RM-weighted decoding? Given that PRM has not been used in the code domain - showing the performance of these baselines is important.
>
> We agree that these baselines are very important. Limited by GPU resources and time, we have only conducted comparisons with the Chain of Thought (CoT) approach so far. We plan to include the additional comparisons you mentioned in the future. Moreover, the inference time for majority voting is significantly greater than our method. For instance, as shown in the results at https://arxiv.org/abs/2305.20050, the PRM approach compared to majority voting only shows significant effects when the sampling number exceeds 100, indicating a substantial inference cost. In contrast, our method's inference overhead is much lower than majority voting. Given the limited work on the use of PRM in code-related problems, we initially focused on the most direct comparison with CoT. We believe that PRM has tremendous potential in code applications and intend to include the baselines you mentioned later to demonstrate the effectiveness of our method.Thank you again for your suggestion and critique, which are instrumental in enhancing the quality of our research.
>
> 2. How does the proposed approach compare to outcome-supervised reward models (ORMs)?
>
> Our mathematical data is based on the dataset available at https://github.com/openai/prm800k. Consequently, we could only train the PRM model for validation. Training the ORM directly on this data would result in a significantly higher number of negative samples compared to positive ones, which is a concern we had to consider. Nevertheless, we intend to
>
> 3. Why is self-assessment more expensive than PRM, given that both the PRM and generator use the same LLM?
>
> This aspect was addressed in our paper. We noted that the complexity of inference for a decoding-only transformer is \( O(N^2) \), where \( N \) represents the sequence length. As the sequence grows, the cost of inference rapidly increases. Therefore, self-assessing the entire sequence is highly resource-intensive. In contrast, PRM fundamentally operates as a classification model and requires only a single forward pass, making it significantly more efficient in this context.